



# Understanding controls on hydrothermal dolomitisation: insights from 3D Reactive Transport Modelling of geothermal convection

Rungroj Benjakul[1], Cathy Hollis[2], Hamish A. Robertson[1], Eric L. Sonnenthal[3], Fiona F. Whitaker[1]

[1]School of Earth Sciences, University of Bristol, Wills Memorial Building, Queens Road, Bristol BS8 1RJ, United Kingdom
[2]School of Earth and Environmental Science, University of Manchester, Manchester M13 9PL, United Kingdom
[3]Energy Geosciences Division, Lawrence Berkeley National Laboratory, Berkeley, CA 94720, USA

*Correspondence to*: Rungroj Benjakul (rb17526@bristol.ac.uk)

**Abstract.**

The dominant paradigm for petrogenesis of high-temperature fault-controlled dolomite, widely known as "hydrothermal
dolomite" (HTD), invokes upwelling of hot fluid along faulted and fractured conduits from a deep over-pressured aquifer. However, this model has several inherent ambiguities with respect to fluid sources and their dolomitisation potential, as well as mechanisms for delivering enough of these reactive fluids to form substantial volumes of dolomite. Here, we use generic 2D and 3D reactive transport simulations of a single transmissive fault system to evaluate an alternative conceptual model whereby dolomitisation is driven by seawater being drawn down into the subsurface and heated. We examine the evolution of
fluid chemistry and distribution of diagenetic alteration, including predictions of the rate, distribution, and temperature of HTD formation, and consider the possible contribution of this process to the Mg-budget of the World's oceans.

The simulations suggest that it is possible for convection of seawater along the fault damage zone to form massive dolomite bodies that extend hundreds of meters vertically and along the fault within a timescale of a few tens of kyr, with no significant alteration of the country rock. Dolomitisation occurs as a gradient reaction by replacement of host limestones and minor
dolomite cementation, and results in discharge of $Mg^{2+}$-poor, $Ca^{2+}$-rich fluids to the sea floor. Fluids sourced from the basement contribute to the transport of heat that is key for overcoming kinetic limitations to dolomitisation, but the entrained seawater provides the $Mg^{2+}$ to drive the reaction. Dolomite fronts are sharper on the "up-flow" margin where $Mg^{2+}$-rich fluids first reach the threshold temperature for dolomitisation, and the "down-flow" dolomite front tends to be broader as the fluid is depleted in $Mg^{2+}$ by prior dolomitisation. The model demonstrates spatial contrasts in the temperature of dolomitisation and the relative
contribution of seawater and basement-derived fluids which are also commonly observed in natural fault-controlled dolomites. In the past, such variations have been interpreted in terms of major shifts in the system driving dolomitisation. Our simulations demonstrate that such changes may also be a product of emergent behaviour within a relatively stable system, with areas that are dolomitised more slowly recording the effect of changes in fluid flow, heat and solute transport that occur in response to diagenetic permeability modification.

Overall, our models robustly demonstrate that high-temperature fault-controlled dolomite bodies can form from mixed convection and act as a sink for Mg in the circulating seawaters. In addition, comparison of our 3D simulations with



simplifications to 2D, indicate that 2D models misrepresent critical aspects of the system. This has important implications for modelling of systems ranging from geothermal resources and mineralisation to carbonate diagenesis, including hydrothermal karstification and ore genesis as well as dolomitisation.

## 1 Introduction

Hydrothermal dolomite (HTD) is commonly thought to form under burial conditions from fluids escaping rapidly from a deep over-pressured aquifer via normal or strike-slip faults, and can take the form of dolomite cement, replacement of limestone and/or overprinting of dolomitic country rocks (Gregg, 2004; Hollis et al., 2017; López-Horgue et al., 2010; Machel, 2004). The economic importance of HTD often lies in the associated hydrothermal dissolution and/or mineralisation, in particular

Mississippi Valley-type Pb-Zn ore deposits (Davies and Smith, 2006; Qing and Mountjoy, 1994). More controversially, fluids thought to have formed fault-related HTDs have also been interpreted as responsible for formation or overprinting of large volumes of matrix-replacement dolomite forming important hydrocarbon reservoirs, inferring basin-wide hydrothermal activity (Al-Aasm, 2003; Davies and Smith, 2006; Sharp et al., 2010; Weissenberger et al., 2006). Machel and Lonnee (2002) were particularly critical of this suggestion and argued strongly against the use of the term hydrothermal (as distinct from

geothermal) without evidence that the dolomites formed at temperatures of at least 5–10 °C higher than the country rock. This would require very rapid rates of both fluid flow and precipitation, and the physics of heat transport mean that it is practically impossible to form HTD away from high permeability conduits. The requisite temperature data to define HTD *sensu stricto* is rarely presented and thus this distinction, whilst technically correct, is often ignored. Rather the term HTD has become widely applied to describe high-temperature fault-related dolomites.

Despite a substantial body of previous work describing HTD geobody distributions and dimensions in outcrop, along with petrographical and geochemical characterisation, important questions remain regarding the process of formation. These include the origin and chemistry of the dolomitising fluids, the volumes of fluids that can be released from deep sources and the mechanisms controlling the distribution and rate of fluid flow, and thus the temperature and rate of dolomitisation. Over-pressured aquifers exist at depth as a result of rapid burial and/or orogenic compression, and tectonic activity can create

permeable pathways for release of such fluids (Ge and Garven, 1994; Oliver, 1986; Whitaker et al., 2004). However, these processes can release, at the most, only a single pore-volume, and thus dolomitisation requires significant focussing of flow from large reservoirs (Consonni et al., 2010; Frazer et al., 2014; Kaufman, 1994). Potential dolomitising fluids are hypothesized as brines or saline connate waters, and in some cases this is supported by high-salinity fluid inclusions in the dolomite cements (Al-Aasm, 2003; Boni et al., 2000; Davies and Smith, 2006; Luczaj et al., 2006). However, a review of the

thermodynamic potential of formation waters to drive dolomitisation (Robertson et al., 2015) suggests fluids hosted in deep aquifers are close to dolomite-calcite equilibrium, and thus have limited dolomitisation potential.





Given the ambiguities in existing conceptual models for HTD formation, here we evaluate an alternative model for formation of high-temperature dolomites by 'top-down' circulation of $Mg^{2+}$-rich fluids from an effectively infinite reservoir of seawater (or modified seawater) overlying a permeable fault damage zone. In high enthalpy areas, heating of groundwater at depth
increases buoyancy and, where the critical Raleigh number is exceeded (Nield, 1968), drives free convection that can entrain water that is cooler and therefore more dense. Controls on this flow system have been analysed numerically by means of 2D models of heat and fluid flow (Person and Garven, 1994; Simms and Garven, 2004), and include sediment and fault permeabilities, spacing between faults, lithologic heterogeneity, and basal heat flow. These studies show convection cells that are characteristically laterally elongated, with an aspect ratio of 1.8, and inflow over broad areas but discharge focussed over
narrower zones. The power of this approach was demonstrated by simulations coupling large-scale free convection of groundwater flow with gravity-driven flow in the Dead Sea rift, Israel (Gvirtzman et al., 1997a, 1997b) that replicate both thermal anomalies and the composition of springs that are mixtures of shallow groundwater with hot brine from deep aquifer.

An increasing number of studies have suggested fault-related dolomitisation by seawater or seawater-derived fluids (Carmichael and Ferry, 2008; Corbella et al., 2014; Haeri-Ardakani et al., 2013; Hollis et al., 2017; Wilson et al., 2007). Thus,
for example in the Hammam Faraun fault (HFF) area, Egypt, strontium isotope and REE signatures suggest that massive dolostone bodies limited to the damage zone of the fault formed from Oligo-Miocene seawater during development of the Suez Rift (Hirani et al., 2018; Hollis et al., 2017). We have previously suggested a conceptual model for formation of these dolostones whereby $Mg^{2+}$-rich seawater could have been drawn down into a convection cell along the fault, heated and ascend to form massive dolostone (Hollis et al., 2017). Testing this hypothesis by reference to the behaviour of modern analogues is
challenging using direct observation of hydrological and chemical processes, and in such scenarios process-based modelling provides an alternative means for rigorous evaluation.

Following the early work of Wilson et al. (2001) and Jones and Xiao (2005), over the past decade process-based numerical modelling of heat and fluid flow coupled with simulation of water-rock reactions using reactive transport models (RTMs) have contributed to understanding of shallow, relatively low-temperature dolomitisation (Al-Helal et al., 2012; Gabellone et al.,
2016; Gabellone and Whitaker, 2016; Garcia-Fresca, 2009; Whitaker and Xiao, 2010; Xiao et al., 2013). RTM simulations of HTD have focussed on understanding controls on the nature of dolomitisation fronts, which are characteristically sharp in fault-related dolomites, and on the development of fault-associated stratabound dolostone bodies (Corbella et al., 2014; Xiao et al., 2013; Yapparova et al., 2017). These studies demonstrate the importance of the geometry of the fault, termination of the fault against a sealing layer and distribution of permeability and precursor mineralogy in the country rock, but are all predicated
on a supply of hot, $Mg^{2+}$-rich fluid at the base of the fault. Consonni et al. (2018) highlight some of the challenges in moving beyond these outcrop-scale models of HTD that arise from uncertainties in fluid composition and volumes, and contrast these with RTM simulation of reflux and thermal convection systems for which fluid flow and chemistry can be better constrained from an understanding of the geometry of the carbonate platform, relative sea-level and climate. Where fluids are derived from compaction, the geometry of the system can help constrain fluid volumes, but specifying fluid chemistries is more challenging



even when the mineralogy of the compacting formation is known (Consonni et al., 2010; Frazer et al., 2014). This points to
the need for a better understanding of deep aquifer and reservoir fluid chemistry, building on the work of Robertson et al.
(2015), to identify controls on dolomitisation potential among different lithological and geological settings.

Simulations of free convection within a magmatic hydrothermal system volcanic caldera system exemplified how faults can
either act as conduits for focussing discharge of high temperature fluids, or for influx of cool shallow fluids from a shallow

reservoir (Jasim et al., 2015). Recently, study of natural fault-controlled dolomite bodies have provided evidence for mixing
of seawater and mantle derived fluids, during dolomitisation, implying that hot, deeply-sourced brines at least provide a thermal
drive for dolomitisation (Breislin et al., in prep.; Koeshidayatullah et al., 2020). The objective of this study is to use RTM to
evaluate whether, for a highly simplified generic single transmissive faulted system, physical and chemical constraints might
allow formation of significant volumes of dolomite by entrainment of seawater into a geothermal convection system. It is not

our objective to reproduce the development of a real dolostone body. Rather, we consider a generic scenario with a simple
geometry under conditions that are clearly defined. The key question addressed is whether the proposed flow system could
account for the volumes and geometries of dolostone geobodies and temperature characteristics observed in fault-related
HTDs, given heat and solute transport, chemical thermodynamics and kinetic constraints. We examine the evolution of fluid
chemistry and distribution of diagenetic alteration, including predictions of the rate, distribution and temperature of dolomite

formation, and the relative contribution of seawater and basement-derived fluids. Additionally, we compare the results of our
3D simulations with those from 2D sections perpendicular to the plane of the fault (echoing this key element of many previous
2D models of HTD), and also 2D models oriented along the plane of the fault.

## 2 Methods

Fluid flow and reactive transport simulations were performed using TOUGHREACT (Sonnenthal et al., 2017; Xu et al., 2011),

which simulates multiphase fluid flow, heat, and solute transport with physical and chemical heterogeneity and incorporates
feedbacks between diagenesis and evolving porosity and permeability (Xu et al., 2004). A number of generic simulations were
run to evaluate the potential of geothermal convection of seawater to form fault-related dolomites within a single transmissive
fault system open to the sea surface. Fluid flow and heat transport was simulated in 2D sections oriented both parallel and
perpendicular to the plane of the fault and compared with that in a 3D domain to constrain an initial condition for the reactive

transport simulations. The 3D domain has a dimension of 2.7 km along the plane of the fault, 2.5 km perpendicular to the fault,
and is 5 km deep (Fig. 1). A symmetry boundary along the centre of the fault damage zone (FDZ) has been assumed, and the
half model is discretised into 4,158 active grid blocks, 150 m wide and 250 m deep, which vary in width from 10 m within the
FDZ to 500 m at the boundaries. The simulated sequence comprises a 3,000 m thick carbonate unit with a permeability of 2 x
$10^{-14}$ m$^2$ ($\approx$ 20 mD) that is sandwiched between two 1,000 m thick lower-permeability ($10^{-15}$ m$^2 \approx$ 1 mD) layers. An 80 m wide

permeable FDZ extends vertically through the carbonate unit and overlying layer to the sea floor. Matrix cells are isotropic,





but within the FDZ vertical permeability is 10 times higher than horizontal permeability and the porosity is 3% higher than that of the aquifer. The properties of all units are listed in Table 1. A full 3D flow model (2.7 x 2.5 x 5 km) has validated the assumed symmetry, and as such the presence or absence of a low permeability fault gouge in the core of the fault would not significantly affect the result.

The two 2D models are based on slices through the 3D model, both perpendicular to the plane of the fault and along the FDZ (Fig. 2). The fault-perpendicular 2D model is 2.6 km wide and includes an 80 m wide FDZ from the surface to a depth of 4 km within a model that extends to a total depth of 5 km. The fault-parallel 2D model represents an 80 m wide FDZ one cell thick and extends 2.7 km laterally and to a depth of 4 km within a model that extends to a total depth of 5 km. Both models are discretized into grid blocks of identical dimensions and with identical rock properties as those in the 3D model, and in all

cases a single porosity–single permeability approach was used.

In all three models, the side boundaries are no-flow, and the bottom and top boundaries are constant temperature and pressure. Infinite volume cells at atmospheric pressure and 25 °C are implemented at the top boundary to represent a shallow overlying body of seawater. The basal cells are at 250 °C, giving an initial conductive temperature gradient of 45 °C km$^{-1}$ and a hydrostatic pressure gradient is imposed. The flux of fluids into the base of the FDZ is determined by the permeability of the

1 km thick basement. Initial parameters and boundary conditions for all model runs are summarised in Fig. 1. The resulting fluid flux is expressed as a Darcy velocity (in m yr$^{-1}$), also known as Darcy flux or specific discharge, and is calculated by multiplying the pore velocity by porosity where all porosity is assumed to be effective. We compare the total mass flow, differentiating seawater and basement fluid drawn into the model, and the discharge of mixed fluid to the sea floor via the 80 m wide FDZ and from the matrix. We model solute transport by diffusion as well as advection, with the same diffusion

coefficient ($10^{-9}$ m$^2$ s$^{-1}$) for all species.

The baseline RTM was constructed using modern open ocean seawater (Nordstrom et al., 1979) as the initial fluid throughout the reservoir and caprock. However, the initial fluid within the basement is at equilibrium with calcite and dolomite and thus has a higher $Ca^{2+}$ concentration and lower $Mg^{2+}$ concentration, alkalinity and pH (Table 2). Geochemical calculations include ten primary aqueous species and two minerals, calcite and dolomite. We use the concentration of $Br^-$ to trace the contribution

of fluids from the basement, hereinafter expressed as basal fluid fraction. The upper 1 km (matrix and FDZ) and basal 1 km (basement) are assumed to be unreactive. Initial mineralogy throughout the interval, 1–4 km depth, is 99 % calcite, with 1 % dolomite as a seed. As the rate of dolomite precipitation is much slower than that of calcite dissolution, calcite was modelled as an equilibrium thermodynamic mineral (fluid and solid equilibrating within a single timestep). Thus, the rate of dolomitisation is determined by the kinetics of dolomite precipitation. The kinetic rate of dolomite precipitation was

constrained by laboratory experiments of Arvidson and Mackenzie (1999) in which dolomite was precipitated at 115–196 °C. We thus include the effects of temperature on both thermodynamics and kinetics of dolomitisation, as well as the evolving fluid composition and mineralogy. The baseline simulation assumes an effective reactive surface area (RSA) of 1,000 cm$^2$ g$^-$



[1], representative of 25-μm-diameter idealized grains (Gabellone and Whitaker, 2016), but does not consider changes in RSA due to diagenetic changes in texture.

Porosity changes in matrix and fractures are directly tied to the volume changes as a result of mineral precipitation and dissolution, and from these the model estimates changes in permeability. TOUGHREACT tracks diagenetic changes in both porosity and permeability that result from changes in mineral volume fractions due to water-rock interaction and subsequently returns this feedback to modify fluid flow. The porosity–permeability relationship for the matrix is approximated by Eq. (1), Carman–Kozeny curve (Bear, 1972) which is one of the simplest and most widely used models for porosity–permeability

relationship (Costa, 2006; Ehrenberg et al., 2006):

$$k = k_i \frac{(1-\phi_i)^2}{(1-\phi)^2} \left(\frac{\phi}{\phi_i}\right)^3 \tag{1}$$

where $k_i$ and $\phi_i$ are the initial permeability and porosity, respectively. Within the fracture zone we represent changes in permeability using the cubic law relation (Steefel and Lasaga, 1994), Eq. (2):

$$k = k_i \left(\frac{\phi}{\phi_i}\right)^3 \tag{2}$$

In an additional 3D simulation we explore the influence of heat flux on temperature, fluid flux, and dolomitisation by reducing the permeability of basement from $10^{-15}$ m$^2$ ($\approx$ 1 mD) to 5 x $10^{-16}$ m$^2$ ($\approx$ 0.5 mD) so limiting the effective flux of hot fluids from the basement, for the same temperature at the base of the model.

Post-processing data analysis includes Mg$^{2+}$-flux calculation and differentiation between dolomite cementation and replacement dolomite. The Mg-flux is calculated as the product of the aqueous concentration of Mg$^{2+}$ (mol kg$^{-1}$) and mass

fluid flux (kg s$^{-1}$ m$^{-2}$) and reported in mmol s$^{-1}$ m$^{-2}$) and provides both a better understanding of source/sink behaviour and a mechanism to track Mg$^{2+}$ circulation and exchange. The changes in total volume of dolomite formed by replacement of calcite (as commonly described by the stoichiometric equation, Eq. (3)) is differentiated from dolomite cementation (primary precipitation of dolomite, Eq. (4)) using the reduction in the volume of calcite and adjusting for the molar volume difference between calcite (36.93 cm$^3$ mol$^{-1}$) and dolomite (64.37 cm$^3$ mol$^{-1}$).

$2CaCO_3 + Mg^{2+} \rightarrow CaMg(CO_3)_2 + Ca^{2+}$ (3)

$Ca^{2+} + Mg^{2+} + 2CO_3^{-2} \rightarrow CaMg(CO_3)_2$ (4)

This ignores the effect of any changes in the volume of calcite that may occur independent of dolomitisation. Because of the retrograde solubility of calcite, this calculation yields a maximum estimate of the fraction of dolomite that forms by replacement in the areas of increasing temperature in the direction of flow and a minimum estimate in areas of decreasing

temperature.



## 3 Results

### 3.1 Baseline 2D and 3D simulations of geothermal convection: fluid flow and heat transfer

The 2D and 3D simulations show circulation of fluids by geothermal convection within the fault damage zone (FDZ) and the combined effects of conduction and advection of heat for a very simple planar fault (Fig. 2). Fluid circulation is driven by

differences in fluid density generated by temperature contrasts between the surface and basement, mediated by the spatial distribution of permeability. The fault-perpendicular 2D model shows the FDZ acting as a conduit for advective discharge of hot fluid sourced largely from the basement, with a vertical Darcy velocity within the FDZ up to 159 m yr$^{-1}$, some 1–2 orders of magnitude higher than rates in the matrix (Fig. 2a, b). Convection and heat transfer between the matrix and the FDZ mainly occur in the aquifer zone (1–4 km depth). A combination of conduction and the ascending flow of buoyant hot fluid from the

basement (average Darcy velocity 1.1 m yr$^{-1}$) elevates the temperature throughout the FDZ (185–246 °C) and in the lower part of the aquifer (235–245 °C at >2.75 km). The total discharge across the 80 m wide FDZ at 30 kyr is 3.44 x 10$^3$ m$^3$ yr$^{-1}$ per m length of fault, equivalent to 9.30 x 10$^6$ m$^3$ yr$^{-1}$ for the entire 2.7 km length of the model. Water from the basement represents 97% of this discharge, as the influx of seawater from the upper boundary through the matrix is limited by the low permeability of the caprock (with Darcy velocity for inflow of only 0.01–0.04 m yr$^{-1}$). and contributes <25% of fluid within the matrix (Fig.

2b, c).

The fault-parallel 2D model shows convection within the FDZ, with flow rates approximately one order of magnitude slower those within the FDZ in the fault-perpendicular 2D model (Fig. 2e, f). Several narrow convection cells develop which draw cold seawater downward into the FDZ across a broad zone. This circulates to the base of the aquifer, with Darcy velocities for the descending limb of up to 6 m yr$^{-1}$, resulting in widespread cooling within the FDZ, and in the ascending limb of up to 16.5

m yr$^{-1}$ at temperatures of up to 72 °C (Fig. 2e, f). The total flow of seawater drawn into the FDZ from the sea along the 2.7 km length of the model is 3.14 x 10$^5$ m$^3$ yr$^{-1}$, with only 15% of this being fluid from the basement. This 2D simulation ignores fluid exchange with the matrix but allows lateral circulation along the plane of the fault gives much lower fluid fluxes and temperatures in the system than the simulation oriented perpendicular to the fault.

The 3D simulation clearly demonstrates the importance of considering both flow along the plane of the FDZ and exchange

with the surrounding matrix (Fig. 2i, j). Fluid flux is strongly focused within the FDZ which accounts for 88% of the total fluid circulation within the system within only 3 % of the total modelled volume. Rates of fluid flow within the FDZ in the 3D simulation are almost one order of magnitude higher than that of the fault-parallel 2D case. Darcy velocities reach a maximum of 280 m yr$^{-1}$ within the FDZ at a depth of 1.2–1.5 km and decrease with depth below this. However, temperature within the FDZ is much higher than in the fault-parallel 2D simulation despite the 3D model predicting an influx of cold seawater within

the FDZ 5 times higher (1.61 x 10$^6$ m$^3$ yr$^{-1}$) than that in the fault-parallel 2D model. This apparent contradiction arises from the inability of the 2D fault-parallel model to account for advective transfer of heat between FDZ and matrix. In the 3D model, seawater at 25 °C that is drawn into the FDZ results in cooling that extends to the top of the basement, and laterally into the





surrounding matrix at a depth of 2.5–4.0 km. High temperature at the base of the model leads to upward flow of fluid through the basement both within the FDZ and, at lower rates, throughout the aquifer (with Darcy velocity of up to 33 m yr$^{-1}$ and 18 m

yr$^{-1}$, respectively). The lower-permeability of the caprock forces ascending fluid within the aquifer matrix back into FDZ at depths shallower than 2.5 km, with a total influx from the upper aquifer to the FDZ of 1.11 x 10$^7$ m$^3$ yr$^{-1}$ at temperatures that range from 115 to 222 °C. This leads to a major contribution of basement water in mixed fluid circulated in most parts of the system, with the exception of the downwelling core (Fig. 2).

### 3.2 Baseline 2D and 3D reactive transport simulations of dolomitisation

The low convective flux of Mg$^{2+}$-rich fluid (seawater) means that, despite the high temperature (>185 °C), dolomitisation in the 2D, fault-perpendicular model is very limited, with replacement of <0.05 volume fraction of the calcite within 30 kyr. Minor dolomitisation occurs where surface-derived fluids enter the reactive aquifer ("D1" in Fig. 2d) and also adjacent to the FDZ at 2.3–3 km depth ("D2" in Fig. 2d). For the 2D fault-parallel model an even smaller amount (<0.005 volume fraction) of dolomite forms at the base of the ascending limbs of the free convection cells close to the basement. Although there is a

high convective flux of Mg$^{2+}$-rich fluid, the associated cooling (<72 °C) means that dolomitisation is limited by sluggish kinetics. This illustrates the importance of having both sufficient flux of Mg$^{2+}$ and temperatures hot enough to overcome kinetic barriers to dolomitisation.

The 3D simulations show how a free convection system developed within the FDZ of a simple planar fault could result in the flux of Mg$^{2+}$-rich seawater across strong thermal gradients and drive dolomitisation. Reactions are largely confined to within

the plane of the fault (Fig. 2l), although convection also drives some exchange of fluids between the FDZ and the matrix. Flow rates within the matrix are 1–2 orders of magnitude less than those within the FDZ, yet flow from the upper part of the aquifer into the fault (at 1–2.1 km depth) is critical to maintaining temperatures within the FDZ that are kinetically favourable to dolomitisation. This "top-down" circulation of seawater drives replacement of calcite by dolomite, reflected in a steep decline in Mg$^{2+}$ along the flow path in areas of active dolomitisation within the FDZ, mirrored by an increase in Ca$^{2+}$ (Fig. 3). The

activity of these ions is a complex function of the effect of water rock interaction and that of mixing between seawater and basement fluid.

Replacement of calcite by dolomite leads to an increase in porosity by up to 7 %, controlled by the difference in mineral density between dolomite and calcite (Fig. 4). Dolomitisation initiates at a depth of 1–2 km in the FDZ, either side of the descending limb of the convection cell, where the temperature exceeds 75 °C. In these areas the precursor calcite is completely replaced

by dolomite within 9 kyr, with a commensurate increase in porosity. Replacement dolomitisation occurs as a gradient reaction, driven by Mg$^{2+}$-rich fluids flowing from cooler to warmer areas. There is minimal alteration of the surrounding matrix, even at very high temperatures (>200 °C in the rising limb of the convection cell), because there is insufficient flux of Mg$^{2+}$-rich fluid. The dolomite front extends laterally over time, reaching a limit of 1 km from the centre of the downwelling limb. At



depths >2 km within the descending limb of the convection cell dolomitisation initially occurs at a slower rate during the
period when dolomitisation "up-flow" at shallower depths consumes $Mg^{2+}$. After 15 kyr, the fluids circulating to depth are less
$Mg^{2+}$-depleted and complete dolomitisation gradually extends to the base of the aquifer. In addition, there is late stage
replacement of calcite in the core of the downwelling limb where low temperatures initially limited reaction rate.

Figure 5 illustrates the importance of temperature and fluid flux in controlling patterns and rates of dolomitisation at contrasting
locations. There are sharp lateral differences in dolomite abundance within the shallow FDZ (1–2 km depth), with a change
from 8 to 70 % dolomite between adjacent grid cells (150 m model resolution). Where dolomite first develops, temperatures
are >100 °C (Location B), and dolomitisation occurs at a rate of up to 10 % $kyr^{-1}$ at simulation times of 3–10 kyr following an
initial 1 kyr induction period of slower dolomitisation. Temperature is slightly higher 0.6 km further along the flow path at
Location C, but here dolomitisation rate is no more than 1.4 % $kyr^{-1}$ within the entire simulation period reflecting the relatively
low $Mg^{2+}$-flux. Dolomitisation occurs most rapidly at location B, reflecting the ample supply of $Mg^{2+}$ from seawater, paired
with a sufficiently high temperature (Fig. 5). Dolomitisation within the downwelling zone at the same depth (Location A) is
very limited for the first 13 kyr, and only initiates after temperatures start to increase in response to the influx of hot waters
from the matrix (which leads to a doubling of heat flux from 13 to 30 kyr). Dolomite abundance decreases more gradually
with depth (Fig. 5), and further insights into the behaviour of the system are provided by comparing locations at different
depths in the downwelling core. Here the higher temperature at depth results in more dolomite forming in the early stages of
the simulation (compare locations A, D and E). However, dolomitisation occurs more slowly at depth, limited by the lower
flux of $Mg^{2+}$.

### 3.3 Role of porosity–permeability feedbacks in controlling patterns of dolomitisation

Most dolomite forms by replacement of calcite, with the associated increase in porosity moderated only slightly by
precipitation of minor amount of dolomite cement. Once all calcite has been replaced, dolomite formation continues but at a
very slow rate (0.015 % $kyr^{-1}$). Dolomite cement represents a minor percentage (<8 %) of the total dolomite at shallow depth
(Locations A and B) and decreases with depth (to 5% at location E) due to the lower $Mg^{2+}$-flux. The baseline 3D simulation
incorporates feedbacks between diagenetically controlled changes in porosity, estimated permeability, heat and solute
transport, and reactions. An increase in porosity of 5.4 % which results from complete replacement of calcite by dolomite
increases the permeability by a factor of two. One impact of this in the upper aquifer is that the later stage dolomitisation
occurs within the core of the descending limb of the convection cells due to the increased transfer of heat from the matrix. In
contrast, in a 3D simulation with no porosity–permeability feedback (Fig. 6) the distribution of temperature and fluid flux do
not evolve with time. The total discharge of hot fluid from the FDZ to the ocean floor amounts to 7.63 x $10^6$ $m^3$ $yr^{-1}$ (reduced
by 9% relative to the baselines simulation with porosity–permeability feedback incorporated). In absolute terms the volume of
recirculated seawater is higher, but this is outweighed by the reduction of fluids from the basement which represent 70% of
the total discharge (compared to 81% in the baseline simulation). The result is that in the simulation in which there is no





increase in permeability driven by replacement dolomitisation (or reduction due to dolomite cements) the FDZ in the shallow aquifer remains undolomitised within the descending limb of the convection cell.

### 3.4 Sensitivity of dolomitisation by geothermal convection of seawater to basal heat flux

Reducing permeability of the basement from $10^{-15}$ m$^2$ ($\approx$ 1 mD) to 5 x $10^{-16}$ m$^2$ ($\approx$ 0.5 mD) significantly reduces the basal heat
flux and fluid flux, with a drop in maximum Darcy velocity within FDZ to 155 m yr$^{-1}$, a rather broader zone of descending flow and a more focussed discharge zone (Fig. 7). The total discharge of hot fluid from the FDZ to the ocean floor is 3.64 x $10^6$ m$^3$ yr$^{-1}$, 44% that of the baseline simulation. The influx of seawater into the FDZ is 15% higher, but there is a substantial (72 %) decrease in total volumetric fluid flux from the basement (Fig. 8). Temperatures in the fault plane are lower due to reduced heat transfer from the basement to the aquifer, and subsequently from the matrix to the fault zone. Temperature within
the FDZ is consequently lower, with a broad (600 m) zone of cool (<80 °C) waters extending to the base of the aquifer and laterally in the FDZ within the lower part of the aquifer. There is also a reduced heat transfer from the basement into the aquifer matrix, and subsequently to the FDZ in the upper part of the aquifer. The result of these differences is a considerably smaller volume of dolomite in the FDZ compared to the 3D baseline simulation, with only partial dolomitisation below 1,750 m depth and a less sharp dolomite front in the zone of ascending flow than seen in the baseline model. There is also a marked difference
in dolomite distribution, with dolomite forming in the ascending limb of the convection cell where temperature exceeds 100 °C, but as in all 3D simulations reactions are constrained to within the FDZ. Dolomitisation occurs at temperatures rather lower than in the baseline simulation, ranging from 155 °C down to a minimum of 73 °C where dolomitisation rates are low.

### 4. Discussion

It has been more than 35 years since Land (1985) recognised that seawater is the only widely available fluid with the capacity
to form extensive bodies of replacement dolomite, although reaction kinetics can limit the rate of dolomitisation at near-surface temperatures (Arvidson and Mackenzie, 1999). Recent geochemical studies (Ryb and Eiler, 2018; Shalev et al., 2019) have presented isotopic data consistent with large-scale dolomitisation by convection of seawater through carbonate sediments. In contrast, over the last decade a major focus of sedimentological research has been to understand the genesis of dolostones developed along extensional or transtensional fault systems. Many of these dolomites form at high temperatures and thus are
generally assumed to be formed from fluids that originate at depth. Release and upward escape of fluids from deep aquifers has become the dominant paradigm (Davies and Smith, 2006), despite considerable debate over the physical and chemical basis for this model and the extent to which it can account for dolomitisation of surrounding limestone (Machel, 2004; Machel and Lonnee, 2002; Robertson et al., 2017).



### 4.1 Comparing RTM predictions with fault-related dolomites

Whilst recognising the simplifications inherent in this narrow subset of possible 3D models, our aim is simply to evaluate the potential of the new conceptual model for fault-related high-temperature dolomitisation and capture the essence of such a system. It is thus important to compare the simulation results with available data describing these dolostones at outcrop. Data extracted from RTM simulations include the dimensions and position of dolomite geobodies within the permeability field, the extent to which alteration extends beyond the FDZ into the matrix and, less commonly, the nature of reaction fronts. Outcrops

provide a snapshot of dolomitisation at the time that reactions ceased, but models can provide time series maps of the development of simulated diagenetic geobodies. Rarely explored is the potential of RTM simulations to provide metrics that relate to the properties of the dolomitising fluids (temperature, salinity, and elemental concentrations/ratios) and can be directly compared with spatial variations in the geochemical and isotopic characteristics of the dolomites. In this example, we have illustrated the further insight offered from tracking changes in temperature through the progress of the reaction at specific

locations, and the difficulties of interpreting evidence of such changes from the rock record alone.

### 4.1.1 Dolomite geobody dimensions

For a FDZ with vertical and along-fault permeability 2 orders of magnitude greater than that of the country rock, our simulations suggest dolomitisation would be constrained almost entirely within the plane of the fault. In the matrix, reactions are limited not by temperature, but by the low flux of $Mg^{2+}$. Note that, *sensu stricto* dolomite in the FDZ could be classified

as hydrofrigid rather than hydrothermal, as the temperatures at which it forms are lower - rather than higher - than that of the country rock (Machel and Lonnee, 2002). As the free convection system draws cool seawater down from the surface, the transfer of heat from the matrix to the FDZ contributes to dolomitisation. We do not here investigate systems with sufficient matrix permeability to give the potential for significant fluid flow between the matrix and FDZ, and rates diffusion are insufficient to allow for dolomitisation. However, even in such scenarios, with a transmissive FDZ providing direct

connectivity between overlying seawater and the geothermal heat source at depth, the drive for dolomitisation of rocks more than a few tens of meters from the FDZ is likely limited. The exception could be where permeable matrix may allow hydraulic continuity between two or more faults, for example across a relay ramp, allowing the development of a single-pass convection system.

The baseline simulation suggests that, for a high basal heat flux, vertically extensive dolomite bodies could form and, over

time, extend laterally, merging to form a single body that might extend for >1 km along the plane of the fault. In comparison, with a lower heat flux, dolomite bodies form at shallower depth and these are more limited vertically and extend only a few hundred metres along the fault. Comparing our generic 3D simulations with the fault-controlled dolostone bodies of the Hammam Faraun Fault described by Hollis et al. (2017) and Hirani et al. (2018) shows marked geometric similarities. The dolostone bodies are constrained to the FDZ of the Hammam Faraun Fault, postdating earlier stratabound dolostones within

Eocene limestone country rock. Whilst the outcrop provides limited information about the vertical extent of the dolostone



bodies (which exceed the 75–80 m height of the outcrop), they extend for up to 500 m along the plane of the fault. Similarly, fault-related dolostones in the Ranero area, northwest Spain that extend >1 km and are defined by sharp contact within the host Aptian-Albian limestone (Dewit et al., 2012; Nader et al., 2012; Shah et al., 2012).

### 4.1.2 Dolomitisation timescales

Our simulations suggest that, for the combination of heat flux and permeability geometry that has been simulated, complete replacement of limestones can occur within the FDZ within the timescale of thousands to a few tens of thousands of years (Fig. 4). The unlimited supply of $Mg^{2+}$ from seawater, coupled with a high geothermal heat flux in tectonically active/rifting settings, enables much more rapid reactions than suggested by previous RTM simulations of fault-controlled diagenesis that suggest timescales ranging from 60 kyr to a few million years, even assuming more favourable kinetics (Abarca et al., 2019;

Consonni et al., 2018; Corbella et al., 2014). Our model simulates a single event lasting a few tens of kyr within a fault that retains its transmissivity, but many outcrop studies suggest multiple episodes of fluids flow, each with the potential to recrystallize the previous dolomite. For example, in the Hammam Faraun example, Sr-isotope dating suggests that the massive dolomite formed over a 10 Ma period but likely within multiple episodes during which fault zone permeability permitted more active circulation (Hirani et al., 2018; Hollis et al., 2017). Fault transmissivity is dynamic, and movement of large quantities

of fluids within faults during earthquakes via "seismic pumping" has been linked to submarine hydrothermal mineralisation that is episodic on time scales ranging from a minutes to tens of thousands of years and controlled by volcanic and tectonic processes (Brantut, 2020; Sibson, 2001; Sibson et al., 1975). Data on permeability changes over longer timescales is sparse (Ingebritsen and Gleeson, 2017), but reconstruction of episodic hydrothermal flow along the Malpais fault in Nevada, suggests either intermittent fluid pulses that lasted ≈1 kyr, or continuous flow that shifted in location every 1 kyr (Louis et al., 2019).

Such timescales are not incompatible with those of dolomitisation suggested by our generic simulations.

### 4.1.3 Characterisation of dolomite fronts:

Our rather simplistic simulations are not designed to evaluate the characteristics of realistic dolomite fronts within the FDZ, given the large size of the model cells both vertically and along the fault, and the associated requirement to use effective permeabilities for cells which are known to be both highly heterogeneous and temporally variable. It would be rewarding to

explore further suggestions from these simulations that at early times, and with low heat flux, large areas of the fault bound geobody remain only partially dolomitised, particularly at depth. The model also raises important questions about contrasts between dolomite fronts in different areas of flow system depending on whether reactions are flux or reaction rate limited. We predict that the character of the dolomite front varies between the margins that receive unmodified seawater (here termed the "up-flow" margins) and those where fluids flow out of the dolomite body ("down-flow margins"). Sharper dolomite fronts

(complete alteration across a single model cell) occur in the "up-flow" dolomite front where dolomitisation is kinetically limited ($Mg^{2+}$-rich fluids first reach the threshold temperature for dolomitisation). In comparison the "down-flow" dolomite front tends to be broader as rates here are flux-limited, as a result of depletion of $Mg^{2+}$ by up-stream dolomitisation. Reactions





fronts perpendicular to the plane of the fault, where model resolution is also higher, are within a single 10 m wide cell. This is in the same order of magnitude as those predicted by 2D simulations of HTDs formed by flow along through-going faults

(Yapparova et al., 2017), and also many field examples with sharp dolomitisation fronts (Grandia et al., 2003; Nurkhanuly and Dix, 2014; Shah et al., 2012). Detailed analysis of dolomitisation fronts within fault-controlled dolomite bodies in nature are remarkably understudied, however, and are likely complex; for example, Koeshidayatullah et al. (2020b) show that reaction fronts can back-step in time, as progressive phases of recrystallization result in porosity reduction (overdolomitisation), restricting the extent to which subsequent fluids can flux away from the fault. An obvious benefit of flow simulations such as

these is that they can provide information on the controls on dolomite body termination during the earliest phases of fluid flux, before porosity–permeability feedback becomes an important control on the extent of fluid migration and reaction.

**4.1.4 Linking RTM output to dolomite geochemistry and isotopic characteristics:**

In addition to evaluating the concept of fault-related dolomitisation by hydrothermal seawater circulation, we hope to illustrate the insights offered a more forensic analysis of simulation data. Specifically, changes in the temperature and chemistry of

fluids during the dolomitisation process can be extracted from RTM output for more meaningful comparison between simulations and paragenetic sequences recorded in the rock record. The contrast between two simple simulations performed at higher heat flux (baseline) and lower heat flux (reduced basement permeability) demonstrates the sensitivity of geobody size and location within the convection cell, but also the robustness of other characteristics.

The temperature of dolomitisation across much of the dolomite body in the baseline simulation is relatively high and tightly

constrained (120–150 °C) at depth (1–2 km). This is slightly higher but comparable with the range for the spatially more restricted dolomite bodies that form in the lower heat flux simulation (100–135 °C), reflecting the dominant role of temperature in controlling reaction rate. However, there are also contrasts between the core and the more marginal areas of the dolomite body, which are clear from the baseline simulation. Within the "up-flow" part of the dolomite body, dolomitisation initially occurs at considerably lower temperatures (80–100 °C) (e.g. locations D and E, Fig. 9), with a step change to higher

temperature as the percentage of dolomite increases. Similarly, in the "down-flow" margins there is a sudden increase in dolomitisation temperature, but here from ≈160 to ≈190 °C (e.g. location C, Fig. 9). These changes occur at the same time, reflecting shifts in patterns of heat and solute transport that occur in response to diagenetic changes in porosity and permeability.

Incorporation of an unreactive tracer (Br⁻) to our RTM allows us to evaluate the relative contribution of basal fluids and

seawater (Fig. 2). Chemical differences in the two fluids (Table 2), together with contrasting boundary water temperatures (250 °C and 25 °C respectively), determine their respective role in formation of the fault-constrained dolomite bodies. Dolomite bodies develop where hydrothermal fluids are mixtures of seawater and basal fluid. Where basal fluids comprise less than half of the fluid, any dolomite that forms in the model does so very slowly. The largest volume of dolomites form from a mixture comprising 59–60% basal fluid and in some areas dolomite formed during the later stages of the simulation records





up to 80% basal fluids. The association between basal-derived fluids, identified from distinctive geochemical and/or isotopic signatures, and high-temperature fault-related dolomite is one commonly observed in the rock record and is generally interpreted to indicate that there is a chemical drive for dolomitisation that is associated with the input of fluid from a deep aquifer (Al-Aasm, 2003; Boni et al., 2000; Davies and Smith, 2006; Luczaj et al., 2006). The RTM clearly calls this assumption into question, as the source of $Mg^{2+}$ for dolomitisation is from the fraction of the fluid derived from the surface seawater.

Instead, the basal fluid controls heat transport within the aquifer and the FDZ by advection, and thus temperature distribution. The basal fluid thus plays a critical role in providing the heat required for relatively rapid formation of dolomite but is not critical to the supply of $Mg^{2+}$ for dolomitisation. The geochemical and isotopic signatures of the resulting dolomites would reflect the component of basement-derived fluids. We might expect, for instance, to see mixing trends between a $^{87}Sr/^{86}Sr$ signature of downwelling fluids reflective of the seawater values, ~ 0.7068–0.70907 (Burke et al., 1982), and more radiogenic

basin-derived fluids. Fluid inclusion and isotope analysis from several studies suggest that the dolomitising fluid might be formed by an evolved seawater or its interaction with rocks and/or other fluids (Gomez-Rivas et al., 2014; Jacquemyn et al., 2014; López-Horgue et al., 2010)

When the porosity–permeability feedback is not implemented in the model, the temperature and basal fluid fraction remain constant through the simulation. With this feedback implemented, the evolution of thermal and chemical conditions for

dolomitisation during reaction progress is a clear example of emergent behaviour, arising from the dynamic nature of the coupled system. However, because of spatial differences in dolomitisation rate, this behaviour would be recorded at different stages within the paragenetic sequence of the dolomites at different locations in a natural dolomite. Simulations thus help to explain complexities in defining a unique paragenetic sequence within some large, natural dolomite bodies

We suggest that these thermal and geochemical trends and are likely to be robust, irrespective of uncertainties in dolomite

kinetics, although assumptions about reaction rates will affect the absolute temperatures (and thus basal fluid fractions) recorded in the simulated dolomites, as will differences in the composition of fluids modelled. Even relatively minor evaporative concentration of the overlying seawater will significantly lower the temperature of dolomitisation (Al-Helal et al., 2012; Gabellone and Whitaker, 2016). Thus, clumped isotope analyses suggesting temperatures <100 °C for the massive dolostone developed in the Hammam Faraun Fault (Hirani et al., 2018) could suggest formation from mesohaline seawater;

not unreasonable as dolomitisation has been dated to the Miocene, when the Gulf of Suez was undergoing desiccation and evaporite precipitation (Hollis et al., 2017). Temperatures predicted from our simple models are more similar to those reported for fault-related dolomites in the Ranero area, northern Spain (120–200 °C; Shah et al., 2010) interpreted to form during multiphase dolomitisation, potentially at variable salinities up to 22 wt.% NaCl (López-Horgue et al., 2010). Numerous other studies combining petrographic, isotopic and fluid inclusion analysis support the suggestion that HTDs with multiple

populations of temperature and fluid composition are likely to be the norm. For example, Luczaj et al. (2006) report fluid inclusion temperatures of 120−150 °C in the HTD of the Devonian Dundee Formation of the Michigan Basin, with an average 21 °C difference between the interior and outer zone of individual dolomite crystals reflecting episodic fault-controlled





transport of fluids and heat and interpreted to originate from the deeper basin. Koeshidayatullah et al. (2020a) demonstrated complex geochemical proxies within the Mount Whyte Formation in the West Canada Sedimentary Basin, suggestive of
mixing between seawater and deep crustal fluids, resulting in dolomite precipitated at high temperature (>200 °C) from very high salinity, metal-enriched fluids at shallow depths. This highlights the need for further work to evaluate the impact of changing composition of the end-member fluids, as well as the effect of solute concentration on density-driven circulation, which in this simulation is driven solely by differences in fluid temperature.

**4.2 Model limitations and new insights from simulations of dolomitisation by hydrothermal convection of seawater**

Systems that are governed by interactions between heat and solute transport and water-rock interaction are challenging to predict or quantify a priori (Ingebritsen et al., 2010). Numerical simulations that represent the coupled nature of these processes can reveal emergent behaviour that is governed by relationships between these separate components. They thus offer a route to evaluate conceptual models of complex systems. However, model results are inevitably affected by the chosen geometry and permeability of the system which controls solute flux and heat transport, and thus the rate and distribution of reactions.

Simulations of structurally complex reservoirs are particularly challenging as they comprise discrete structures with properties that contrast strongly with those of the country rock and range over vastly different length scales (Matthäi et al., 2007). Our models do not attempt to capture the complexities of flow within FDZs, for which the porous media formulation employed by TOUGHREACT is inappropriate. Nor do we consider how the permeability of the FDZ may change due to geomechanical responses to variations in the stress field that are localised in time and space, or the physical and chemical effects of changes
in state of the fluid, for example due to boiling. Rather we simply demonstrate that, where major discontinuities provide high permeability pathways for flow, it is possible in high enthalpy settings for interaction between heat supplied from the basement and $Mg^{2+}$-rich seawaters from above to form high-temperature fault-bound dolomite bodies. The magnitude and anisotropy of permeability, as well as the effective reactivity, of the country rock will also influence the extent to which this fault-associated convection system could form dolomites in the adjacent matrix, or alternatively might drive recrystallisation of a dolomite
precursor formed during earlier diagenesis.

Although this model deals solely with seawater and an equivalent salinity basal fluid, there is a clear opportunity now to evaluate the reactions that occur within convective systems involving basal fluids of different geochemical characteristics and densities, as well as investigating alternative surface water compositions. More complex mineral assemblages should also be investigated, both in terms of host-rock mineralogy and diagenetic minerals that may form. Our simple simulations
incorporating the effect of replacement dolomitisation on permeability demonstrate the importance of better understanding the impact of the full range of both physical and chemical controls on the evolving permeability. They also offer three substantive learnings with implications that extend beyond the contentious realm of genesis of HTD.





Firstly, although our generic models are very simple, they robustly demonstrate that, under the conditions simulated, high temperature fault-controlled dolostone bodies can form by seawater convection. $Mg^{2+}$-rich seawater is drawn downward into
the permeable damage zone around a fault that extends to shallow depth, mixing with hot buoyant fluids escaping from the basement. Dolomitisation occurs within the FDZ as a gradient reaction, with replacement of host limestones and minor dolomite cementation as circulating fluids flow across isotherms. Fluids from the basement play an important role in transporting heat into the FDZ, but the $Mg^{2+}$ to drive dolomitisation comes from the entrained seawater even though this is <50% and locally <20% of the fluid within the FDZ. Our reactive transport simulations thus reconcile the apparently
contradictory observations that on one hand identify geochemical and isotopic signatures for basement fluids in many high-temperature dolomites, and on the other indicate that almost all basement fluids are depleted in $Mg^{2+}$ and thus do not have the potential for dolomitisation.

Secondly, comparison of our 3D baseline model with simplifications in 2D, both perpendicular and parallel to the plane of the FDZ, illustrate how 2D models fail to represent critical aspects of system behaviour (Fig. 2a–h). The alignment of convective
flow with fault planes has long been recognised (López and Smith, 1995, 1996; Murphy, 1979) and is supported by recent 3D thermal models of the Upper Rhine Graben (Armandine Les Landes et al., 2019) that show good agreement between simulated and predicted thermal anomalies. Our RTM simulations show that 2D models oriented perpendicular to the fault cannot represent the convection that occurs along the plane of the fault within the permeable damage zone, which is key to generating massive fault-constrained dolostone bodies. It is likely that prior simulations using such 2D models oriented perpendicular to
the plane of the fault (Corbella et al., 2014; Frixa et al., 2014; Jones and Xiao, 2013; Xiao et al., 2013) misrepresent these important aspects of reactions driven by circulation of fluid along the fault plane. They might not therefore well explain the diagenetic evolution of fault-guided diagenetic reactions, specifically overestimating the influence of fault-controlled fluid convection control on the adjacent matrix. Circulation within the FDZ is seen in the 2D model oriented along the plane of the fault (Fig. 2e–h). However, because this model fails to account for the effect of heat transfer between the matrix and FDZ, in
this case the system appears to have been too cool for significant dolomitisation. Our clear conclusion is that numerical analysis of systems where faults impact permeability must be conducted in 3D, and this has important implications for modelling a range of systems, from geothermal resources and mineralisation fluid to carbonate diagenesis, including dolomitisation and hydrothermal karstification.

Finally, dolomitisation associated with hydrothermal and geothermal circulation of seawater has been suggested as an
important but poorly constrained sink for Mg in seawater (Shalev et al., 2019). These processes thus have fundamental implications for understanding the global budgets of Mg, Ca and C, which are linked by the common controls of weathering, volcanism and carbonate precipitation (Arvidson et al., 2011; Elderfield, 2010; Holland, 2005). Combining recent magnesium isotope ($^{26}Mg$) data for low temperature hydrothermal fluids with evidence of a constant $^{26}Mg$ over the past 20 Myr, Shalev et al. (2019) suggest that there is a significant unexplained flux of Mg from the modern ocean (estimated at 1.5–2.9 Tmol $yr^{-1}$).
Mg mass balance calculations for our simple generic models suggest that the dolomitisation that results from this mixed





convection system within permeable fault zones may provide a sink for 2.0–2.6 x10⁷ mol Mg yr⁻¹ for each km of fault. At such rates, in order to account for all the cryptic dolomite identified by Mg-isotope studies, such a system would need to be established along total fault length of 60–150 x 10³ km. For comparison, the East African Rift system extends some 6.4 x 10³
km, with extension accommodated by numerous intra-rift faults in addition to those bordering the rift (Shillington et al., 2020
and references therein). This suggests Mg drawdown by fault-related dolomitisation in areas of very high localised heat flux may be significant, but is likely to be subsidiary to dolomitisation driven by geothermal ("Kohout") convection within platform carbonates. The absence of permeable conduits or high basal heat flux mean much lower rates of dolomitisation in these systems (Whitaker and Xiao, 2010), but this is amply compensated by the vast area of major carbonate platform systems globally (currently some 800,000 km² of low latitude oceans, Milliman, 1993). Further resolution of these fluxes is key to
critically assessing the role of dolomitisation relative to weathering and seafloor spreading in determining secular changes in the Mg/Ca of global ocean (Hardie, 1996; Holland and Zimmermann, 2000; Shalev et al., 2019).

## 5. Conclusions

Hydrothermal dolomitisation has been synonymous with dolomitisation driven by release of over-pressured fluids from depth, but our simulations offer another potential interpretation. An alternative concept considers that high-temperature dolomite
could replace a precursor limestone where seawater is drawn down into a convection system that is developed within a steep high permeability conduit during periods of high heat flux. Such a scenario may be most likely developed in extensional basins, when faults breach the seafloor and geothermal gradients are elevated. Evaluating this concept using 2D and 3D reactive transport simulations shows that when a deep-seated permeable fault extends to the sea floor, the influx of high $Mg^{2+}/Ca^{2+}$ seawaters can drive replacement of calcite by dolomite with only minor dolomite cementation. The rate of advective transfer
of heat by basal-derived fluids relative to that of cold seawater from above will determine the distribution of dolomite within the flow field and size of the dolomite geobodies that form within the convective cell. This new model provides one solution to the $Mg^{2+}$ mass balance problems inherent in HTD formation by release of hot over-pressured fluids from a deep reservoir. It also may explain at least some of the missing $Mg^{2+}$ from the World's oceans. The new insights and key conclusions from our study include:

• RTM simulations suggest that convection of that leads to a mixture of hot basal fluids and seawater along the FDZ can form sizeable ($10^2$–$10^3$ m) fault-bound dolomite bodies within timescales of thousands of years in high enthalpy settings.
   • Reactions are largely confined to within the FDZ and dolomitisation occurs at temperatures of 80–160 °C and from fluids that comprise a significant (>0.5, and locally >0.8) fraction of basal-derived fluids.





• Modification of porosity and permeability during diagenesis can lead to systematic changes in the temperature and fraction of basal fluid in the hydrothermal mixture as progress of the reaction that may be recorded in the paragenetic sequence.

• A permeable fault damage zone provides a corridor along which fluids will preferentially flow. As this is not represented in 2D simulations perpendicular to the plane of the fault such model construction is fundamentally

inappropriate and could lead to misleading suggestions for example about the extent of associated alteration of the country rock.

• Heat transfer between the fault zone and country rock plays an important role in contributing to dolomitisation within the fault damage zone which cannot be observed using 2D simulations along the plane of the fault.

3D RTM simulations can enhance understanding of the mechanisms and controls on dolomitisation within faulted and fractured

systems, and the geometry and spatial distribution of the resulting high-temperature dolomite geobodies. RTMs offer insights into temporal evolution of reactions, and an opportunity to examine not only the geometry of diagenetic geobodies and the nature of reaction fronts, but also a broader range of metrics that can be directly related to isotopic and geochemical characteristics of diagenetic products. Critically, numerical analysis of systems where faults impact permeability must be conducted in 3D, and this has important implications for modelling of systems ranging from geothermal resources to ore

formation and carbonate diagenesis, including hydrothermal karstification as well as dolomitisation.

**Author contribution**

F.F.W. and R.B. wrote the manuscript. R.B. and F.F.W. also designed and developed the conceptual model and experiments. Model construction, simulations and analyses were conducted by R.B. under supervision of F.F.W.. E.L.S. provided technical support with the simulator-TOUGHREACT. H.R. advised on initial model construction. C.H., E.L.S and H.R. gave advice,

feedback, and comments on the manuscript.

**Competing interests.**

The authors declare that they have no conflict of interest.

**Acknowledgements**

Rungroj Benjakul is supported by a scholarship from Chiang Mai University and this work was partially supported by a joint

industry project, PD3 (Prediction of Deposition, Deformation and Diagenesis in Carbonates).



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





**Table 1. Hydrological and thermal properties of units in models. Porosity is fractional and permeability is isotropic in all units except for the fault damage zone. For the 3D model we also evaluate sensitivity to a 50% lower basement permeability. Permeability in SI**
**units (m²) can be converted to approximate Darcies by multiplying by $10^{12}$.**

| Lithological units | Porosity | Permeability | | | Grain Density | Grain Specific | Thermal | Tortuosity |
|---|---|---|---|---|---|---|---|---|
| | $\phi$ | $k_x$ | $k_y$ | $k_z$ | $\rho$ | heat capacity | conductivity | |
| | | (m²) | (m²) | (m²) | (kg m⁻³) | (J kg⁻¹ K⁻¹) | (W m⁻¹ K⁻¹) | |
| Caprock | 0.20 | 1E-15 | 1E-15 | 1E-15 | 2,120 | 1,000 | 3.72 | 0.2 |
| Aquifer | 0.20 | 2E-14 | 2E-14 | 2E-14 | 2,300 | 1,000 | 3.72 | 0.2 |
| Basement | 0.01 | 1E-15 | 1E-15 | 1E-15 | 2,300 | 1,000 | 3.72 | 0.2 |
| Lower-permeability basement | 0.01 | 5E-16 | 5E-16 | 5E-16 | 2,300 | 1,000 | 3.72 | 0.2 |
| Fault damage zone (FDZ) | 0.21 | 2E-12 | 2E-13 | 2E-12 | 2,000 | 1,000 | 3.33 | 0.2 |





**Table 2. Chemical composition of end-member fluids in reactive transport analysis. Primary aqueous species concentrations are in mol kg$^{-1}$ of H$_2$O; Mineral saturation indices are log(IAP/K). Br$^-$ is used as a non-reactive tracer of the contribution of basal fluid.**

| Components | Modern seawater | Basal water |
|---|---|---|
| pH | 8.22 | 6.49 |
| Na$^+$ | 4.854E-01 | 4.854E-01 |
| Ca$^{2+}$ | 1.066E-02 | 5.862E-02 |
| Mg$^{2+}$ | 5.507E-02 | 5.288E-03 |
| K$^+$ | 1.058E-02 | 1.058E-02 |
| Cl$^-$ | 5.657E-01 | 5.657E-01 |
| SO$_4$$^{2-}$ | 2.926E-02 | 2.926E-02 |
| HCO$_3$$^-$ | 2.406E-03 | 5.863E-04 |
| Br$^-$ | 1.000E-07 | 1.000E-04 |
| Mg$^{2+}$/Ca$^{2+}$ molar ratio | 5.17 | 0.09 |
| Calcite Saturation Index | +0.59 | 0.00 |
| Dolomite Saturation Index | +2.00 | 0.00 |





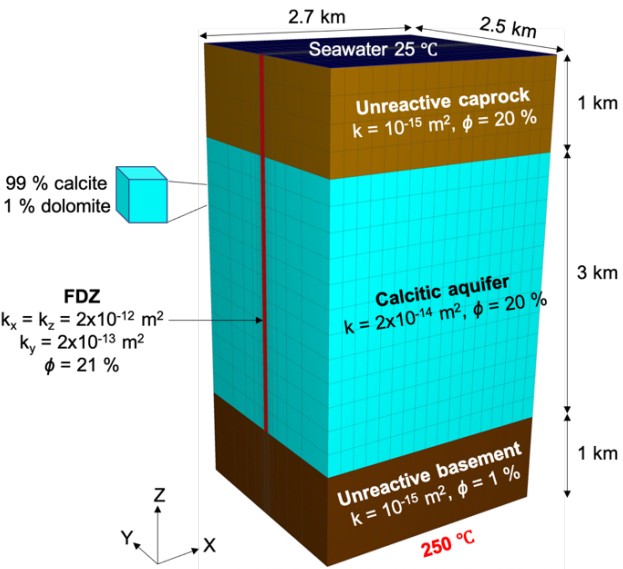

**Figure 1. Model dimensions, material properties, and simulation grid. The permeable fault damage zone (FDZ) is displayed in red. The top boundary is open to exchange with overlying seawater at 25 °C and the base of the model is set at 250 °C, giving an initial geothermal gradient of 45 °C km⁻¹. Parallel to the FDZ the grid is regularly discretized with dimension of 150 x 250 m (width x height), but perpendicular to the fault cells thickness varies from 500 m at the edge of the model to 10 m within the FDZ. The cap rock and basement are unreactive, and the aquifer is calcitic, with 1% "seed" dolomite.**



**Figure 2.** Simulation results for 2D fault-perpendicular (a–d) and fault-parallel (e–h), and baseline 3D (i–l) models showing heat transfer, fluid flow (with representative streamlines), basal fluid fraction, and dolomite distribution at 30 kyr. Darcy velocities are positive when flow is upward and negative when flow is downward. Note the yellow planes (2D) and volume (3D) on the left of each row represent the orientation of 2D and 3D displays, and the white dashed line in (l) highlights the area of significant dolomitisation for which the temporal evolution of the system is presented in Fig. 4.







**Figure 3. Temporal evolution of dolomite volume fraction (a), Mg$^{2+}$ (b), and Ca$^{2+}$ concentration (c) from the 3D baseline simulation and plots of those changes down the core of the descending limb of the convection cell within the FDZ (white dashed line). Ca$^{2+}$ is enriched relative to seawater and Mg$^{2+}$ is depleted as a result of replacement of precursor limestone by dolomite. After complete replacement, there is a minor decrease in concentrations of both Mg$^{2+}$ and Ca$^{2+}$ due to dolomite cementation.**


**Figure 4. Temporal evolution of temperature, Mg²⁺-flux (fluid flux x Mg²⁺ concentration), dolomite abundance (volume %), and change from initial porosity (volume %) along the fault plane for the 3D baseline simulation after 5, 10, 15, 20 and 30 kyr. Two separate dolomite geobodies start to form in the FDZ in the upper part of the aquifer, with large areas where more than half the calcite has been replaced by dolomite. Over time dolomitisation these merge to a single massive dolomite body with sharp dolomitisation fronts. The downwelling core is initially a zone of minor calcite cementation, but the dominance of replacement dolomite over dolomite cementation results in an increase of porosity of up to 7 %.**





Figure 5. Dolomitisation, Mg²⁺-flux, temperature and porosity change in the FDZ, at locations vertically along the descending flow path (A–D–E) and laterally with distance from the core (A–B–C). Note how replacement dolomitisation is initially limited by the low temperature in the "up-flow" margin of the dolomite body (A), whilst in the "down-flow" margin (C) reactions are limited by Mg²⁺-flux.



**Figure 6. Development of the dolomite geobody within the FDZ cutting along middle of the fault plane for the 3D simulation (a) with porosity–permeability feedback turned off and (b) the baseline 3D simulation that includes feedback between porosity and permeability that results in localised advection of heat from the matrix and dolomitisation in the downwelling core (b).**




**Figure 7.** Simulation results at 30 kyr from (a) baseline 3D model and (b) lower permeability basement model showing heat transfer and fluid flow (greyscale arrows) within the FDZ and distribution of dolomitisation. Darcy velocities are positive when flow is upward and negative when flow is downward. The lower heat flux with a less permeable basement results in cooler conditions, lower fluid flux, and two smaller dolomite bodies form in zones of upwelling rather than a single large dolomite body in the zone of descending flow.





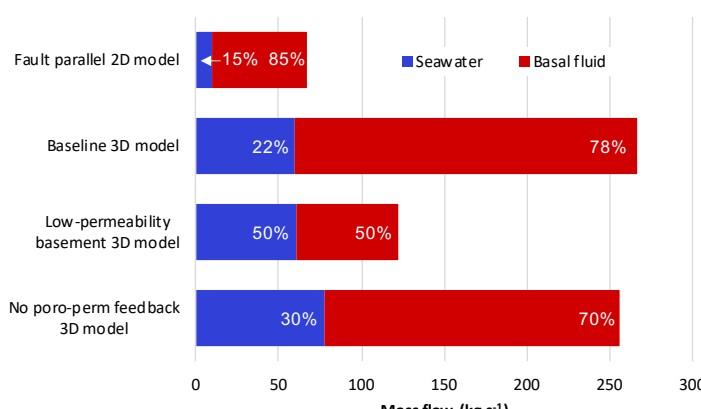

**Figure 8. Total discharge to the sea floor from simulations of geothermal convection within the FDZ, differentiating contributions of seawater drawn down into the fault-damage zone (FDZ) and basal fluid.**



**Figure 9. (a) Temperature of dolomitising fluid during reaction progress at selected locations in the FDZ in the baseline 3D simulation, with distance laterally away from downwelling core (A–B–C) and vertically through the core (A–D–E). Locations are shown on the inset overlain on the temperature distribution at 15 kyr. Circles represent 5 kyr intervals, with a cross to mark 15 kyr, midway through the simulation. (b) Basal fluid fraction of dolomitising fluid during reaction progress at locations A–E in the FDZ in the baseline 3D simulation. The contour lines, 0.1, 0.5, and 0.9 represent the abundance of dolomite in mineral fraction. Dolomites record lower temperatures and basal fluid fraction in the core of the downwelling limb and higher temperatures and basal fluid fraction towards the upwelling limb of the convection cell. Locations where dolomite forms more slowly also show a step change in temperature with the addition of more warm water derived from the basement and delivered via the matrix as dolomitisation increases permeability in the FDZ.**