# Peer review of "Understanding controls on hydrothermal dolomitisation: insights from 3D Reactive Transport Modelling of geothermal convection"

_Solid Earth, 2020_

## Referee Comment (RC1) · Merce Corbella (Referee) · 15 Jul 2020

The manuscript entitled "Understanding controls on hydrothermal dolomitization: insights from 3D RTM of geothermal convection" presents very interesting numerical simulation exercises that give light to the problem of dolomite formation and Mg budgets. The exercises constitute very simplified scenarios of an ample fault system cross-cutting from surface to a calcitic aquifer and basement, where the comparison between 2D and 3D models is of high significance. Although the results obtained are only descriptive of natural systems with the exact model settings, the authors rightly extrapolate some of the findings to general controls for hydrothermal dolomitization. The main

contribution of the manuscript is to illustrate an alternative model of high temperature dolomitization where seawater provides the Mg and a basement fluid provides the heat for the chemical reaction to take place. The text is well structured and clear. The Introduction to the problem of hydrothermal dolomitization is very comprehensive. Similarly, the Methodology section is very detailed. Overall, this is a great preprint to read, well written and illustrated. However, there are some issues that could be considered to improve the scientific significance of the results and to make it easier for the reader to fully understand the models.

a. A justification of caprock porosity and permeability, as well as basement permeability would help in accepting the simplified model as realistic. The values used seem to be rather high.

b. Also, an explanation for the chemical composition of the basement fluid would be nice, as a quite alkaline and dilute fluid is being used for the simulations, which do not correspond to a usual warm and acidic brine.

c. Figure 1 illustrates very well most of the parameters of the system simulations. However, consider adding a figure 1b with the initial and boundary conditions of the simulations (temperature, fluxes, etc) so that we can easily visualize their influence over the simulation results.

d. The modelled system has a tall prismatic shape, which, together with the no-flow boundaries used, forces convection to form narrow vertical cells. I'm afraid this also constraints the shape and size of the resulting dolostone bodies.

e. Check figures 2b, f and j, which show vertical Darcy velocity fields, for the arrows superimposed. The arrows must correctly reflect the flow directions but are quite confusing where they appear to 'crash' (cap rock flow with convection in aquifer, basement flow with convective flow). Consider using flow lines or small velocity vectors instead.

f. The 3D model results of Fig 2 can hardly be observed in the plane perpendicular

to the fault. The temperature variations, velocity field or dolomite formed cannot be appreciated there. Consider adding a 2D cross-cut of it or modifying the orientation of the 3D prism or enlarging the figure.

g. A thorough sensitivity analysis is important. I would like to see the results of simulations with lower basal temperature, an acidic brine as basal fluid, smaller permeabilities for caprock and basement and a much wider system.

Besides, here is a list of technical corrections that should also be taken into account.

1.Order the citations to references in the same sentence time-wise. Normally, the cites are organized from older to more recent. E.g. Introduction, lines 35-40, Discussion lines 350, 480, etc).

2. Consider not to include references that cannot be found by many researchers, such as Breislin et al., Robertson et al. 2015.

3.Correct the reference Almandine Les Landes et al. 2019, line 565: there's no volume number.

4.Some references are missing from the list, like Gòmez-Rivas et al. 2014.

5.First sentence of paragraph in lines 420: spare 'and'.

6.In paragraph of lines 245, is "mineral density" appropriate, or should it rather be 'molar volume"?

---

## Referee Comment (RC2) · Taury Smith (Referee) · 3 Aug 2020

This is a very good paper that should be published. I am going to lay out the hydrothermal dolomitization model that Graham Davies and I proposed in 2006 and compare it to this modeling study in hopes that the authors might incorporate some of these ideas. I really like the idea of mixing of seawater with hydrothermal fluids and I proposed a similar model back in 2008 in an abstract for a talk I gave at AAPG Eastern Section that won the best talk award at that meeting. So there is clearly support for this concept! I sent the abstract to Fiona.

In the model we proposed, hydrothermal dolomitization was thought to mainly occur

at relatively shallow burial depths of <500m. Most faults associated with hydrothermal dolomite reservoirs die out within or just above the dolomitized zone suggesting very early faulting at relatively shallow depths. There are commonly seals such as black shales or anhydrite immediately overlying carbonates that have undergone hydrothermal alteration. The Bowling Green Fault Zone in Ohio, one of the great HTD reservoirs in the world, was never buried more than 1km even at its maximum burial. The faults are almost always transtensional faults which would have extremely high permeability (probably multiple orders of magnitude greater than the matrix. We think, as this paper suggests, that the matrix permeability of the host controls the distance from the fault conduit that dolomitization might occur. Most alteration occurs when the faults are actively moving rather than in in a stationary fault. Fault movement might drive "seismic pumping" that would draw fluids up from depth, and perhaps down from the seawater source as well. Commonly, but not always, the most hydrothermal alteration is concentrated to the uppermost permeable unit below the seal, perhaps in zones only 10s of meters thick.

The shallow depth of alteration would fit very well with the mixing of seawater and hydrothermal fluids proposed in this paper. Perhaps the fault moves, there is an episode of high pressure, high temperature fluid flow up the fault and that fluid then mixes with whatever fluid is currently residing in the formation, probably seawater or slightly modified seawater. Burial depths of 50-500m would only help this model as it is that much closer to the seafloor. It would be nice to see some additional stratigraphy in the model that showed variations in permeability and alteration closer to the surface.

I have some animations that I will send to Fiona as well that illustrate some of these concepts.

All that said, this is great work and a great contribution and it should be published.

Taury Smith

---

## Author Comment (AC2)

**Author comment to RC1 (Merce Corbella)**

We thank the reviewer for carefully reading our manuscript and providing us with valuable feedback for improving the manuscript. We copy below the reviewer comments (in italic) and a point-by-point response (in plain text).

a. A justification of caprock porosity and permeability, as well as basement permeability would help in accepting the simplified model as realistic. The values used seem to be rather high.

**Response to a:** We acknowledge the reviewer's comment. As now explained in the Methods section in the revised manuscript we use "caprock" and "basement" as generic terms indicating layers of lower permeability relative to the reactive carbonate aquifer which are overlying and underlying the aquifer respectively and behave effectively as semi-confining units. As our model simply demonstrates hydrothermal dolomitisation by seawater convection within a generic faulted heterogeneous system, not a specific case or area, we deliberately keep the variation in rock properties to a minimum. The initial fractional porosity (0.2) and permeability of the caprock (1 mD) represent fractured rock or semi-consolidated sediment where the porosity is quite high due to the lesser effect of compaction and diagenetic alteration. The critical value is that for permeability, which at 1 mD is comparable to values assigned for a shallow shaley layer overlying the carbonate reservoir of the Barremian Toca Fm in the Lower Congo Basin (Consonni et al., 2018). Although the porosity of 0.2 for caprock is quite high, several prior studies show that the influence of changing porosity on the flow field and convection cell patterns is negligible (Gow et al., 2002; Kühn et al., 2006; Zhao et al., 2003). However, it is still important to note that porosity affects the difference between the average pore velocity and the Darcy velocity.

The reviewer is correct that in some situations the shallow permeability may be lower and we now include the results of additional simulations with smaller caprock permeabilities of 0.5 mD and 0.1 mD. These suggest that within the range investigated this parameter has very little effect on the behavious of the system relative to the base case. To further investigate the sensitivity to basement permeability in controlling heat and fluid flux from deep sources, we have also included an additional simulation with lower basement permeability i.e. 0.1 mD within the revised manuscript (section 3.4).

b. Also, an explanation for the chemical composition of the basement fluid would be nice, as a quite alkaline and dilute fluid is being used for the simulations, which do not correspond to a usual warm and acidic brine.

**Response to b:** Basement fluid is high-temperature modern seawater at equilibrium with calcite and dolomite and thus has a higher  $Ca^{2+}$  concentration, lower  $Mg^{2+}$  concentration and alkalinity, and slightly acidic. The basement fluid has been explained in the Methods section (line 153–155) and Table 2 in the revised manuscript. Further simulations of alternate basement fluid chemistries are the subject of ongoing investigations.

c. Figure 1 illustrates very well most of the parameters of the system simulations. However, consider adding a figure 1b with the initial and boundary conditions of the simulations (temperature, fluxes, etc) so that we can easily visualize their influence over the simulation results.

**Response to c:** We have modified figure 1, adding an initial plot (figure 1b) to show the initial and boundary conditions. No boundary flux has been imposed in this system to avoid an uncertainty from specifying the flux. Rather fluxes across the boundaries are determined by density contrasts and permeability enabling us to show the effect of how these evolve over time.

Figure 1. (a) Model dimensions, material properties, and simulation grid. The permeable fault damage zone (FDZ) is displayed in red. Parallel to the FDZ the grid is regularly discretized with dimension of 150 x 250 m (width x height), but perpendicular to the fault cells thickness varies from 500 m at the edge of the model to 10 m within the FDZ. The caprock and basement are unreactive, and the aquifer is calcitic, with 1% "seed" dolomite. The side boundaries are no-flow, and the top and bottom boundaries are constant temperature. (b) The initial conditions of the simulations. Top boundary is open to exchange with overlying seawater at 25 °C and the base of the model is set at 200 and 250 °C, giving an initial geothermal gradient of 35 and 45 °C km-1, respectively.

*d.* The modelled system has a tall prismatic shape, which, together with the no-flow boundaries used, forces convection to form narrow vertical cells. I'm afraid this also constraints the shape and size of the resulting dolostone bodies.

**Response to d:** The dimensions of the model are one of a number of uncertainties in 3D RTM simulation of complex HTD system, which also include boundary conditions and petrophysical properties as discussed above. Although we agree that the shape and size of dolostone bodies could be influenced by our model setting, model width is only one of the minor controls on fluid circulation. These are already well explained by the classic studies such as those of Bethke (1989). In real systems the permeability structure within the fault damage zone will be complex distribution and will likely dominate the pattern of hydrothermal convection (Guillou-Frottier et al., 2020; Harcouët-Menou et al., 2009). We thank the reviewers for raising this point and have incorporated statements in the second paragraph of section 4.2 in the revised manuscript which clarifies these issues.

Again, our main point of this study is to demonstrate that the proposed mechanism is a viable means of forming HTD, and can explain the apparent incongruity between geological observations (in particular fluid inclusion temperatures and salinities of the dolomites) and challenges in satisfying Mg2+ mass balance requirements worthy of consideration. We hope

this paper will act as a jumping-off point for further studies applying this concept to specific outcrop or subsurface case studies.

- e. Check figures 2b, f and j, which show vertical Darcy velocity fields, for the arrows superimposed. The arrows must correctly reflect the flow directions but are quite confusing where they appear to 'crash' (cap rock flow with convection in aquifer, basement flow with convective flow). Consider using flow lines or small velocity vectors instead.
- f. The 3D model results of Fig 2 can hardly be observed in the plane perpendicular to the fault. The temperature variations, velocity field or dolomite formed cannot be appreciated there. Consider adding a 2D cross-cut of it or modifying the orientation of the 3D prism or enlarging the figure.

**Response to e and f:** We have modified Figure 2 by using velocity vectors of each individual cell to represent flow direction. The 3D diagrams are revised in Figure 3 for a better presentation with enlarged 3D solids.

---

## Author Comment (AC3)

**Author comment to RC1 (Merce Corbella)**

We thank the reviewer for carefully reading our manuscript and providing us with valuable feedback for improving the manuscript. We copy below the reviewer comments (in italic) and a point-by-point response (in plain text).

a. *A justification of caprock porosity and permeability, as well as basement permeability would help in accepting the simplified model as realistic. The values used seem to be rather high.*

**Response to a:** We acknowledge the reviewer's comment. As now explained in the Methods section in the revised manuscript we use "caprock" and "basement" as generic terms indicating layers of lower permeability relative to the reactive carbonate aquifer which are overlying and underlying the aquifer respectively and behave effectively as semi-confining units. As our model simply demonstrates hydrothermal dolomitisation by seawater convection within a generic faulted heterogeneous system, not a specific case or area, we deliberately keep the variation in rock properties to a minimum. The initial fractional porosity (0.2) and permeability of the caprock (1 mD) represent fractured rock or semi-consolidated sediment where the porosity is quite high due to the lesser effect of compaction and diagenetic alteration. The critical value is that for permeability, which at 1 mD is comparable to values assigned for a shallow shaley layer overlying the carbonate reservoir of the Barremian Toca Fm in the Lower Congo Basin (Consonni et al., 2018). Although the porosity of 0.2 for caprock is quite high, several prior studies show that the influence of changing porosity on the flow field and convection cell patterns is negligible (Gow et al., 2002; Kühn et al., 2006; Zhao et al., 2003). However, it is still important to note that porosity affects the difference between the average pore velocity and the Darcy velocity.

The reviewer is correct that in some situations the shallow permeability may be lower and we now include the results of additional simulations with smaller caprock permeabilities of 0.5 mD and 0.1 mD. These suggest that within the range investigated this parameter has very little effect on the behavious of the system relative to the base case. To further investigate the sensitivity to basement permeability in controlling heat and fluid flux from deep sources, we have also included an additional simulation with lower basement permeability i.e. 0.1 mD within the revised manuscript (section 3.4).

b. *Also, an explanation for the chemical composition of the basement fluid would be nice, as a quite alkaline and dilute fluid is being used for the simulations, which do not correspond to a usual warm and acidic brine.*

**Response to b:** Basement fluid is high-temperature modern seawater at equilibrium with calcite and dolomite and thus has a higher $Ca^{2+}$ concentration, lower $Mg^{2+}$ concentration and alkalinity, and slightly acidic. The basement fluid has been explained in the Methods section (line 153–155) and Table 2 in the revised manuscript. Further simulations of alternate basement fluid chemistries are the subject of ongoing investigations.

c. *Figure 1 illustrates very well most of the parameters of the system simulations. However, consider adding a figure 1b with the initial and boundary conditions of the simulations (temperature, fluxes, etc) so that we can easily visualize their influence over the simulation results.*

**Response to c:** We have modified figure 1, adding an initial plot (figure 1b) to show the initial and boundary conditions. No boundary flux has been imposed in this system to avoid an uncertainty from specifying the flux. Rather fluxes across the boundaries are determined by density contrasts and permeability enabling us to show the effect of how these evolve over time.

[Figure]

Figure 1. (a) Model dimensions, material properties, and simulation grid. The permeable fault damage zone (FDZ) is displayed in red. Parallel to the FDZ the grid is regularly discretized with dimension of 150 x 250 m (width x height), but perpendicular to the fault cells thickness varies from 500 m at the edge of the model to 10 m within the FDZ. The caprock and basement are unreactive, and the aquifer is calcitic, with 1% "seed" dolomite. The side boundaries are no-flow, and the top and bottom boundaries are constant temperature. (b) The initial conditions of the simulations. Top boundary is open to exchange with overlying seawater at 25 °C and the base of the model is set at 200 and 250 °C, giving an initial geothermal gradient of 35 and 45 °C km$^{-1}$, respectively.

*d. The modelled system has a tall prismatic shape, which, together with the no-flow boundaries used, forces convection to form narrow vertical cells. I'm afraid this also constraints the shape and size of the resulting dolostone bodies.*

**Response to d:** The dimensions of the model are one of a number of uncertainties in 3D RTM simulation of complex HTD system, which also include boundary conditions and petrophysical properties as discussed above. Although we agree that the shape and size of dolostone bodies could be influenced by our model setting, model width is only one of the minor controls on fluid circulation. These are already well explained by the classic studies such as those of Bethke (1989). In real systems the permeability structure within the fault damage zone will be complex distribution and will likely dominate the pattern of hydrothermal convection (Guillou-Frottier et al., 2020; Harcouët-Menou et al., 2009). We thank the reviewers for raising this point and have incorporated statements in the second paragraph of section 4.2 in the revised manuscript which clarifies these issues.

Again, our main point of this study is to demonstrate that the proposed mechanism is a viable means of forming HTD, and can explain the apparent incongruity between geological observations (in particular fluid inclusion temperatures and salinities of the dolomites) and challenges in satisfying Mg$^{2+}$ mass balance requirements worthy of consideration. We hope

this paper will act as a jumping-off point for further studies applying this concept to specific outcrop or subsurface case studies.

e. *Check figures 2b, f and j, which show vertical Darcy velocity fields, for the arrows superimposed. The arrows must correctly reflect the flow directions but are quite confusing where they appear to 'crash' (cap rock flow with convection in aquifer, basement flow with convective flow). Consider using flow lines or small velocity vectors instead.*
f. *The 3D model results of Fig 2 can hardly be observed in the plane perpendicular to the fault. The temperature variations, velocity field or dolomite formed cannot be appreciated there. Consider adding a 2D cross-cut of it or modifying the orientation of the 3D prism or enlarging the figure.*

**Response to e and f:** We have modified Figure 2 by using velocity vectors of each individual cell to represent flow direction. The 3D diagrams are revised in Figure 3 for a better presentation with enlarged 3D solids.

[Figure]

Figure 2. Simulation results for 2D fault-perpendicular (a–d) and fault-parallel (e–h) models showing heat transfer, fluid flow (with representative streamlines), basal fluid fraction, and dolomite distribution at 30 kyr. The yellow planes on the left of each row represent the orientation of 2D displays and the red outlines highlight the FDZ. Darcy velocities are positive when flow is upward and negative when flow is downward. Note the difference in scale for Darcy velocity between the 2D fault-perpendicular and fault-parallel models.

[Figure]

Figure 3. Three-dimensional baseline simulation illustrating heat transfer, fluid flow, basal fluid fraction, and dolomite distribution at 30 kyr. Note Darcy velocities are positive when flow is upward and negative when flow is downward. The 3D volume on the left represent the orientation and dimensions of the displays. The dolomitisation is focussed along the plane of the fault which the temporal evolution of the system is presented in Fig. 5.

*g. A thorough sensitivity analysis is important. I would like to see the results of simulations with lower basal temperature, an acidic brine as basal fluid, smaller permeabilities for caprock and basement and a much wider system.*

**Response to g:** These points are addressed in responses to reviewer's comments a and b above. With reference to the lower basal temperature, we have included an additional simulation illustrating sensitivity to this parameter by reducing the temperature at the base of the model from 250 to 200 °C. We also now include further simulations investigating (separately) the influence of a lower permeability of basement (0.1 mD) and caprock (0.5 mD and 0.1 mD) in the revised manuscript. However, a simulation of much wider system is omitted but explained above in our response to comment d and also discussed in the revised manuscript. We agree that the fluid chemistries are interesting and alternative investigations of more acidic brines are underway as part of a subsequent study.

However, we would like to clarify that the main purpose of this manuscript is a generic evaluation of the new conceptual model of hydrothermal dolomitisation by top-down

circulation of seawater, which is successfully demonstrated by our 3D RTM simulations and represents a significant advance on those for HTD previously published which have been exclusively 2D perpendicular to the plane of the fault and simulate the response to injection of fluids at specified locations and rates. As mentioned in section 4.2 (Model limitations) in the manuscript, a more fully developed sensitivity analysis on rock properties, fluid compositions, model dimension etc. are important as areas for further work.

**List of technical corrections**

*1. Order the citations to references in the same sentence time-wise. Normally, the cites are organized from older to more recent. E.g. Introduction, lines 35-40, Discussion lines 350, 480, etc).*

**Response to 1:** We acknowledge the reviewer's comment. However, the order of in-text citations herein is based on relevance and alphabetical listing with response to the manuscript preparation guideline.

*2. Consider not to include references that cannot be found by many researchers, such as Breislin et al., Robertson et al. 2015.*

**Response to 2:** We have updated the reference of Breislin et al. which now is accepted by the Journal of Sedimentary Research, though it is currently still in press. However, we decline to remove the study of Robertson et al., 2015 as it is one of key studies pointing out a limited dolomitisation potential of fluids in deep aquifers. The abstract of this work can be accessed via the following link:
https://www.cspg.org/common/Uploaded%20files/pdfs/documents/conference_website/carbonates/Abstract_BookV2.pdf

*3. Correct the reference Almandine Les Landes et al. 2019, line 565: there's no volume number.*
4. *Some references are missing from the list, like Gòmez-Rivas et al. 2014.*

**Response to 3 and 4:** We have crossed checked all references and corrected that.

*5. First sentence of paragraph in lines 420: spare 'and'.*

**Response to 5:** Sorry, we cannot find that.

*6. In paragraph of lines 245, is "mineral density" appropriate, or should it rather be 'molar volume"?*

**Response to 6:** Yes, it should be molar volume. We have replaced the word "mineral density" by "molar volume" in the revised manuscript.

**References:**

Bethke, C. M.: Modeling subsurface flow in sedimentary basins, Geol Rundsch, 78(1), 129–154, doi:10.1007/BF01988357, 1989.

Breislin, C., Crowley, S. F., Banks, V., Marshall, J., Millar, I., Riding, J. and Hollis, C.: Controls on dolomitization within extensional basins: an example from the Derbyshire Platform, UK, J. Sediment. Res., in press.

Consonni, A., Frixa, A. and Maragliulo, C.: Hydrothermal dolomitization: simulation by reaction transport modelling, J. Geol. Soc. London Spec. Publ., 435(1), 235–244, doi:10.1144/SP435.13, 2018.

Gow, P. A., Upton, P., Zhao, C. and Hill, K. C.: Copper-gold mineralisation in New Guinea: Numerical modelling of collision, fluid flow and intrusion-related hydrothermal systems, Aust. J. Earth Sci., 49(4), 753–771, doi:10.1046/j.1440-0952.2002.00945.x, 2002.

Guillou-Frottier, L., Duwiquet, H., Launay, G., Taillefer, A., Roche, V. and Link, G.: On the morphology and amplitude of 2D and 3D thermal anomalies induced by buoyancy-driven flow within and around fault zones, Solid Earth, 11(4), 1571–1595, doi:https://doi.org/10.5194/se-11-1571-2020, 2020.

Harcouët-Menou, V., Guillou-Frottier, L., Bonneville, A., Adler, P. M. and Mourzenko, V.: Hydrothermal convection in and around mineralized fault zones: insights from two- and three-dimensional numerical modeling applied to the Ashanti belt, Ghana, Geofluids, 9(2), 116–137, doi:10.1111/j.1468-8123.2009.00247.x, 2009.

Kühn, M., Dobert, F. and Gessner, K.: Numerical investigation of the effect of heterogeneous permeability distributions on free convection in the hydrothermal system at Mount Isa, Australia, Earth Planet. Sci. Lett., 244(3), 655–671, doi:10.1016/j.epsl.2006.02.041, 2006.

Robertson, H., Whitaker, F. F., Palmer, T. and Gabellone, T.: The Search for Fluids with HTD Potential, CSPG-SEPM Mountjoy carbonate meeting, Banff, Alberta, 23-28 August 2015, 2015

Zhao, C., Hobbs, B. E., Mühlhaus, H. B., Ord, A. and Lin, G.: Convective instability of 3-D fluid-saturated geological fault zones heated from below, Geophys. J. Int., 155(1), 213–220, doi:10.1046/j.1365-246X.2003.02032.x, 2003.

---

## Author Response (AR1)

Dear Solid Earth Editor,

We are grateful for the time spent by you and the reviewers on our manuscript. As already indicated in our comments in the interactive discussion, we have revised the manuscript by implementing the reviewer suggestions, making the message and figures clearer and more compelling. Please find below the point-by-point responses to the reviewer comments (in *italic*) and relevant changes made in the manuscript, as well as the original Word file annotated with changes tracked so that you can easily see the changes that were made in response to the reviewer comments.

We believe that we have taken care of all of the reviewer comments, and hope that you will find the manuscript now acceptable for publication in Solid Earth.

We look forward to hearing from you soon.

Sincerely,

Rungroj Benjakul, 14 October 2020

**Author comment to RC1 (Merce Corbella)**

*a. A justification of caprock porosity and permeability, as well as basement permeability would help in accepting the simplified model as realistic. The values used seem to be rather high.*

**Response to a:** We acknowledge the reviewer's comment. As now explained in the Methods section in the revised manuscript we use "caprock" and "basement" as generic terms indicating layers of lower permeability relative to the reactive carbonate aquifer which are overlying and underlying the aquifer respectively and behave effectively as semi-confining units. As our model simply demonstrates hydrothermal dolomitisation by seawater convection within a generic faulted heterogeneous system, not a specific case or area, we deliberately keep the variation in rock properties to a minimum. The initial fractional porosity (0.2) and permeability of the caprock (1 mD) represent fractured rock or semi-consolidated sediment where the porosity is quite high due to the lesser effect of compaction and diagenetic alteration. The critical value is that for permeability, which at 1 mD is comparable to values (for example) assigned for a shallow shaley layer overlying the carbonate reservoir of the Barremian Toca Fm in the Lower Congo Basin (Consonni et al., 2018). Although a porosity of 0.2 for caprock is quite high, several prior studies show that the influence of changing porosity on the flow field and convection cell patterns is negligible (Gow et al., 2002; Kühn et al., 2006; Zhao et al., 2003). However, it is still important to note that porosity affects the difference between the average pore velocity and the Darcy velocity as mentioned in lines 146–148.

The reviewer is correct that in some situations the shallow permeability may be lower and we now include the results of additional simulations with smaller caprock permeabilities of 0.5 mD and 0.1 mD. These suggest that, within the range investigated, this parameter has very little effect on the behaviour of the system relative to the base case. To further investigate the sensitivity to basement permeability in controlling heat and fluid flux from deep sources, we have also included an additional simulation with lower basement permeability i.e. 0.1 mD within the revised manuscript (section 3.4).

**Changes in manuscript:** As explained above, the issue with caprock and basement properties has been explored in the revised manuscript (lines 128–130 and 481–485) and the sensitivity analysis to lower permeabilities of basement (lines 176–178 and 295–312) and caprock (lines 182–184 and 320–323) have been included in the revised manuscript. Additional relevant references have also been added (lines 659–661, 714–716 and 819–820).

b. *Also, an explanation for the chemical composition of the basement fluid would be nice, as a quite alkaline and dilute fluid is being used for the simulations, which do not correspond to a usual warm and acidic brine.*

**Response to b:** Basement fluid is high-temperature modern seawater at equilibrium with calcite and dolomite and thus has a higher $Ca^{2+}$ concentration, lower $Mg^{2+}$ concentration and alkalinity, and slightly acidic. The basement fluid has been explained in the Methods section (lines 153–155) and Table 2 in the revised manuscript. Further simulations of alternate basement fluid chemistries are the subject of ongoing investigations.

**Changes in manuscript:** we have modified an explanation of the basement fluid in lines 153–155.

c. *Figure 1 illustrates very well most of the parameters of the system simulations. However, consider adding a figure 1b with the initial and boundary conditions of the simulations (temperature, fluxes, etc) so that we can easily visualize their influence over the simulation results.*

**Response to c:** We have modified figure 1, adding a new plot (figure 1b) to show the initial and boundary conditions. No boundary flux has been imposed in this system to avoid an uncertainty from specifying the flux. Rather fluxes across the boundaries are determined by density contrasts and permeability enabling us to show the effect of how these evolve over time.

**Changes in manuscript:** Figure 1 (page 29) has been modified to include the boundary condition (Fig. 1a) and initial condition (Fig. 1b).

[Figure]

Figure 1. (a) Model dimensions, material properties, and simulation grid. The permeable fault damage zone (FDZ) is displayed in red. Parallel to the FDZ the grid is regularly discretized with dimension of 150 x 250 m (width x height), but perpendicular to the fault cells thickness varies from 500 m at the edge of the model to 10 m within the FDZ. The caprock and basement are unreactive, and the aquifer is calcitic, with 1% "seed" dolomite. The side boundaries are no-flow, and the top and bottom boundaries are constant temperature. (b) The initial conditions of the simulations. Top boundary is open to exchange with overlying seawater at 25 °C and the base of the model is set at 200 and 250 °C, giving an initial geothermal gradient of 35 and 45 °C km$^{-1}$, respectively.

> d. *The modelled system has a tall prismatic shape, which, together with the no-flow boundaries used, forces convection to form narrow vertical cells. I'm afraid this also constraints the shape and size of the resulting dolostone bodies.*

**Response to d:** The dimensions of the model are one of a number of controlling variables in 3D RTM simulation of this HTD system, which also include boundary conditions and petrophysical properties (discussed above). Although we agree that the shape and size of dolostone bodies could be influenced by our model setting, model width is only one of the controls on the aspect ratio of the geothermal cells. These are already well explained by the classic studies such as those of Bethke (1989). In real systems the permeability structure within the fault damage zone will be complex distribution and will likely dominate the pattern of hydrothermal convection (Guillou-Frottier et al., 2020; Harcouët-Menou et al., 2009). We thank the reviewers for raising this point and have incorporated statements in the second paragraph of section 4.2 in the revised manuscript which clarifies these issues.

Again, our main point of this study is to demonstrate that the proposed mechanism is a viable means of forming HTD, and can explain the apparent incongruity between geological observations (in particular fluid inclusion temperatures and salinities of the dolomites) and challenges in satisfying Mg$^{2+}$ mass balance requirements. We hope this paper will act as a jumping-off point for further studies applying this concept to specific outcrop or subsurface case studies.

**Changes in manuscript:** We have clarified this issue by stating the explanation above in the second paragraph of section 4.2 in the revised manuscript (lines 479–492) and additional relevant references have been added (lines 667–669 and 678–680).

*e. Check figures 2b, f and j, which show vertical Darcy velocity fields, for the arrows superimposed. The arrows must correctly reflect the flow directions but are quite confusing where they appear to 'crash' (cap rock flow with convection in aquifer, basement flow with convective flow). Consider using flow lines or small velocity vectors instead.*

*f. The 3D model results of Fig 2 can hardly be observed in the plane perpendicular to the fault. The temperature variations, velocity field or dolomite formed cannot be appreciated there. Consider adding a 2D cross-cut of it or modifying the orientation of the 3D prism or enlarging the figure.*

**Response to e and f:** We have modified Figure 2 by using velocity vectors of each individual cell to represent flow direction. The 3D diagrams are revised in Figure 3 for a better presentation with solid colour in each cell representing the values at the cell centres rather than continuous contours generated numerically.

**Changes in manuscript:** The 3D model results have been removed from Figure 2 (page 30) and restructured in Figure 3 (page 31) showing values in each cell rather than contoured extrapolations from these, and better perspective view. All simulation results from TOUGHREACT are also regenerated with a python code "trexplot" developed by co-author (Hamish Robertson) to visualise cell-centred blocks rather than interpolated fields. Small velocity vectors have been utilized to represent flow direction.

[Figure]

Figure 2. Simulation results for 2D fault-perpendicular (a–d) and fault-parallel (e–h) models showing heat transfer, fluid flow (with representative streamlines), basal fluid fraction, and dolomite distribution at 30 kyr. The yellow planes on the left of each row represent the orientation of 2D displays and the red outlines highlight the FDZ. Darcy velocities are positive when flow is upward and negative when flow is downward. Note the difference in scale for Darcy velocity between the 2D fault-perpendicular and fault-parallel models.

[Figure]

Figure 3. Three-dimensional baseline simulation illustrating heat transfer, fluid flow, basal fluid fraction, and dolomite distribution at 30 kyr. Note Darcy velocities are positive when flow is upward and negative when flow is downward. The 3D volume on the left represent the orientation and dimensions of the displays. The dolomitisation is focussed along the plane of the fault which the temporal evolution of the system is presented in Fig. 5.

*g. A thorough sensitivity analysis is important. I would like to see the results of simulations with lower basal temperature, an acidic brine as basal fluid, smaller permeabilities for caprock and basement and a much wider system.*

**Response to g:** These points are addressed in responses to reviewer's comments a and b above. With reference to the lower basal temperature, we have included an additional simulation illustrating sensitivity to this parameter by reducing the temperature at the base of the model from 250 to 200 °C. We also now include further simulations investigating (individually) the influence of a lower permeability of basement (0.1 mD) and caprock (0.5 mD and 0.1 mD) in the revised manuscript. A simulation of much wider system is omitted but explained above in our response to comment d and also discussed in the revised manuscript. We agree that the fluid chemistries are interesting and alternative investigations of more acidic brines are underway as part of a subsequent study.

However, we would like to clarify that the main purpose of this manuscript is a generic evaluation of the new conceptual model of hydrothermal dolomitisation by top-down circulation of seawater, which is successfully demonstrated by our 3D RTM simulations and represents a significant advance on those for HTD previously published which have been exclusively 2D perpendicular to the plane of the fault and simulate the response to injection of fluids at externally specified locations and rates. As mentioned in section 4.2 (Model limitations) in the manuscript, a more fully developed sensitivity analysis on rock properties, fluid compositions, model dimension etc. are important as areas for further work.

**Changes in manuscript:** Additional sensitivity analysis to lower basal temperature, and smaller permeabilities for caprock and basement have been included in the revised manuscript. These relevant modifications can be found in the Method section (lines 176–183), section 3.4 (lines 294–323), Table 1 (page 27), Figure 8 (page 36), Figure 9 (page 37), and Figure 10 (page 38).

**List of technical corrections**

*1. Order the citations to references in the same sentence time-wise. Normally, the cites are organized from older to more recent. E.g. Introduction, lines 35-40, Discussion lines 350, 480, etc).*

**Response to 1:** We acknowledge the reviewer's comment. However, the order of in-text citations herein is based on relevance and alphabetical listing with response to the manuscript preparation guideline.

**Changes in manuscript:** None

*2. Consider not to include references that cannot be found by many researchers, such as Breislin et al., Robertson et al. 2015.*

**Response to 2:** We have updated the reference of Breislin et al. which now is accepted by the Journal of Sedimentary Research, though it is currently still in press. However, we would prefer not to remove reference to the study of Robertson et al., 2015 as it is one of key studies pointing out a limited dolomitisation potential of fluids in deep aquifers. The abstract of this work can be accessed via the following link:
https://www.cspg.org/common/Uploaded%20files/pdfs/documents/conference_website/carbonates/Abstract_BookV2.pdf

**Changes in manuscript:** The references have been updated (lines 102, 614–615, and 760–761).

*3. Correct the reference Almandine Les Landes et al. 2019, line 565: there's no volume number.*
*4. Some references are missing from the list, like Gòmez-Rivas et al. 2014.*

**Response to 3 and 4:** We have crossed checked all references and corrected that.

**Changes in manuscript:** The reference of Almandine Les Landes et al. (2019) has been corrected with the volume number "2019" (lines 598–600) while that of Gòmez-Rivas et al. (2014) has been added (lines 656–658).

*5. First sentence of paragraph in lines 420: spare 'and'.*

**Response to 5:** Sorry, we cannot find that.

**Changes in manuscript:** None

*6. In paragraph of lines 245, is "mineral density" appropriate, or should it rather be 'molar volume"?*

**Response to 6:** Yes, it should be molar volume. We have replaced the word "mineral density" by "molar volume" in the revised manuscript.

**Changes in manuscript:** The word "mineral density" has been replaced by "molar volume" (line 253)

**Response:** We welcome the positive feedback on our manuscript and are glad that someone with a strong rock-based background and such extensive experience of HTD at outcrop and in the subsurface found the manuscript easy to read and coherent with wide range field examples on which they have worked.

We agree that fluid mixing is key here and have referred to the Smith et al. (2008) abstract in both our Introduction and Discussion sections. However, it is our view that, hydrothermal dolomitisation in Trenton and Black River Groups would more likely occur from mixing of hydrothermal fluids with large volumes of seawater drawn down from the sea floor (as proposed here), rather than by mixing associated with displacement of original pore fluids, where the volumes of seawater (and thus the mass of $Mg^{2+}$ available) would likely be several orders of magnitude too small to explain the volumes of HTD described.

The observations made by the reviewer about the early faulting at shallow depth (<500 m) are indeed in line with the model we propose. It may be that hydrothermal alteration of the overlying low permeability shales / anhydrite has obscured evidence of tectonically-related high permeability features that connected the HTD bodies to the sea floor. We agree that these transtensional fault zones (note there is no vertical offset across our simple modelled fault) would have a very significant permeability contrast with the carbonate matrix.

**Changes in manuscript:** Revised manuscript makes reference to the suggested work and the observations from the literature are added in support of our study (lines 102–106, 363–365, and 459–460). Additional relevant references have also been added (lines 767–769 and 789–790)

*It would be nice to see some additional stratigraphy in the model that showed variations in permeability and alteration closer to the surface.*

**Response:** Our ongoing work incorporates additional complexity not just in the matrix but also within the fault damage zone, and support the observation of greater alteration at shallow depth. Many challenges remain in understanding HTD and we agree it will be useful to simulate specific case studies, although challenges of parameterisation, in particular of 3D variations in permeability in both space and time are considerable. We will certainly include your suggestion coupled with a fully systematic sensitivity analysis in the future work.

**Changes in manuscript:** None

[revised manuscript text omitted]

---

## Author Response (AR2)

Dear Executive Editor,

Thank you for your consideration and timely response regarding the revised submission. Following your advice, we have removed the reference Robertson et al. (2015) and added a word of acknowledgement to reviewers. The changes we have made can be found in marked-up manuscript below.

Sincerely,

Rungroj Benjakul, 30 October 2020

[revised manuscript text omitted]